# Codelivery of HBx-siRNA and Plasmid Encoding IL-12 for Inhibition of Hepatitis B Virus and Reactivation of Antiviral Immunity

**DOI:** 10.3390/pharmaceutics14071439

**Published:** 2022-07-09

**Authors:** Yan Mu, Xiao-He Ren, Di Han, Ying-Ying Guan, Pei-Ling Liu, Si-Xue Cheng, Hong Liu

**Affiliations:** 1National & Local Joint Engineering Research Center of High-Throughput Drug Screening Technology, College of Life Sciences, Hubei University, Wuhan 430062, China; 201911110711158@stu.hubu.edu.cn (Y.M.); 201811110711016@stu.hubu.edu.cn (Y.-Y.G.); 202121107012457@stu.hubu.edu.cn (P.-L.L.); 2Key Laboratory of Biotechnology of Chinese Traditional Medicine of Hubei Province, College of Life Sciences, Hubei University, Wuhan 430062, China; 3Hubei Collaborative Innovation Center for Green Transformation of Bio-Resources, College of Life Sciences, Hubei University, Wuhan 430062, China; 4Key Laboratory of Biomedical Polymers of Ministry of Education, Department of Chemistry, Wuhan University, Wuhan 430072, China; 2017202030134@whu.edu.cn (X.-H.R.); 2020102030108@whu.edu.cn (D.H.); chengsixue@whu.edu.cn (S.-X.C.)

**Keywords:** hepatitis B treatment, siRNA, plasmid DNA, immune regulation, macrophage polarity

## Abstract

Chronic hepatitis B is a critical cause of many serious liver diseases such as hepatocellular carcinoma (HCC). The main challenges in hepatitis B treatment include the rebound of hepatitis B virus (HBV)-related antigen levels after drug withdrawal and the immunosuppression caused by the virus. Herein, we demonstrate that the HBV-related antigen can be effectively inhibited and antiviral immunity can be successfully reactivated through codelivery of the small interfering RNA (siRNA) targeting HBV X protein (HBx) and the plasmid encoding interleukin 12 (pIL-12) to hepatocytes and immune cells. After being treated by the siRNA/pIL-12 codelivery system, HBx mRNA and hepatitis B surface antigen (HBsAg) are dramatically reduced in HepG2.215 cells. More importantly, the downregulated CD47 and programmed death ligand 1 (PD-L1) and the upregulated interferon-β promoter stimulator-1 (IPS-1), retinoic acid-inducible gene-1 (RIG-1), CD80, and human leukocyte antigen-1 (HLA-1) in treated HepG2.215 cells indicate that the immunosuppression is reversed by the codelivery system. Furthermore, the codelivery system results in inhibition of extracellular regulated protein kinases (ERK) and phosphoinositide-3-kinase (PI3K)/protein kinase B (Akt) pathways, as well as downregulation of B-cell lymphoma-2 (Bcl-2) and upregulation of p53, implying its potential in preventing the progression of HBV-induced HCC. In addition, J774A.1 macrophages treated by the codelivery system were successfully differentiated into the M1 phenotype and expressed enhanced cytokines with anti-hepatitis B effects such as interleukin 6 (IL-6) and tumor necrosis factor-α (TNF-α). Therefore, we believe that codelivery of siRNA and pIL-12 can effectively inhibit hepatitis B virus, reverse virus-induced immunosuppression, reactivate antiviral immunity, and hinder the progression of HBV-induced hepatocellular carcinoma. This investigation provides a promising approach for the synergistic treatment of HBV infection.

## 1. Introduction

Chronic hepatitis B is a major cause of many serious liver diseases, causing serious public health problems [1,2]. Currently, the commonly used drugs for the treatment of chronic hepatitis B are nucleoside analogs and interferon alpha (IFN-α). Nucleoside analogs require lifelong medication, while IFN-α is only effective for some patients [3,4]. The critical issues in the treatment of chronic hepatitis B include clearance of cccDNA and genomic HBV DNA, reduction of antigen load, and reactivation of exhausted immune responses in the hepatic microenvironment [5]. To achieve these goals, different types of new anti-HBV drugs are under development, such as siRNA [6], virus entry inhibitors [7], nucleocapsid protein inhibitors [8,9], and therapeutic vaccines [10,11].

As a potential type of anti-hepatitis B therapeutic agent, siRNA is able to inhibit mRNA during the transcription of HBV DNA, thus reducing the corresponding antigen load. In this study, we selected HBx as the target of siRNA silencing. HBx plays a critical role in HBV replication by binding with the nuclear cccDNA mini chromosome to initiate and maintain viral transcription [12]. Furthermore, a previous report demonstrated that inhibiting HBx using an NQO1 inhibitor could block cccDNA transcription [13]. In addition, some studies have shown that HBx is correlated with the development of HBV-induced hepatocellular carcinoma [14,15,16,17]. Therefore, inhibition of HBx is of great significance for anti-HBV therapy and prevention of liver cancer caused by HBV. Nevertheless, siRNA does not directly activate immunity. In order to achieve an effective inhibition of HBV after drug withdrawal, it is necessary to activate immunity to enhance the therapeutic efficiency.

Up to now, diverse strategies for activating immunity to inhibit HBV have been reported. For example, GS-9620, a Toll-like receptor 7 agonist, exhibits anti-hepatitis B effects by stimulating peripheral blood mononuclear cells to produce IFN-α [18]. A study on different phenotypes of IFN-α showed that IFN-α14 is the most effective subtype [19].

As it is well known, macrophages can differentiate into two phenotypes, i.e., the M1 type secreting proinflammatory cytokines (e.g., TNF-α and IL-6) and the M2 type secreting anti-inflammatory cytokines (e.g., IL-10) [20]. HBV can promote macrophages to differentiate into the M2 phenotype and inhibit differentiation into the M1 phenotype, thus inducing immune tolerance to allow HBV replication in the host [21,22,23,24].

With the ability to induce production of interferon-γ (IFN-γ), promote the differentiation of T helper 1 (TH1) cells, and link innate immunity with adaptive immunity [25,26], IL-12 shows promising anti-hepatitis B effects, which have been verified in animal models and clinical practice [27,28]. In addition, IL-12 can promote macrophages to differentiate into the M1 phenotype [29]. However, the detailed mechanism of inhibition of HBV replication by IL-12 is not yet understood well enough.

Compared with single therapeutic modes, synergistic therapy can improve the efficiency of anti-hepatitis B. For instance, combination therapy based on Crispr/Cas9 targeting HBV-PreS1 and PD-1 results in upregulated CD80, CD83, CD86, IL-6, IL-12, IL-23, and TNF-α and favors immune activation to inhibit HBV more effectively [30]. More effective HBsAg inhibition, correlated with more efficient induction of innate immune responses, can be achieved by combination therapy based on ARB-1740, a capsid inhibitor, and pegylated IFN-α [31].

Although a variety of synergistic therapy approaches have been reported, there is no related research on the synergistic treatment of chronic hepatitis B using a combination of siRNA targeting HBx and plasmid encoding IL-12 (pIL-12). In view of the promising antiviral effects of siRNA and favorable immune regulation capability of IL-12, a codelivery system of an siRNA targeting HBx and plasmid encoding IL-12 was prepared for effective inhibition of HBV and reactivation of antiviral immunity to optimize the therapeutic actions and improve prognosis. The anti-hepatitis B and immunomodulatory effects of the codelivery system on hepatocytes (HepG2.215 cells) and immune cells (J774A.1 macrophages) were studied. The codelivery system can effectively inhibit HBV, reverse the immunosuppression caused by HBV, prevent the progression of HBV-induced hepatocellular carcinoma, and modulate macrophages toward the antiviral M1 phenotype. This investigation provides a promising strategy to improve the therapeutics and prognosis of chronic hepatitis B.

## 2. Materials and Methods

### 2.1. Reagents

Lipofectamine 2000 (lipo) and pUNO1-mIL12 encoding IL-12 (pIL-12) were obtained from Invitrogen. siRNA targeting HBx (sense: 5′AACGACCGACCUUGA- GGCAUATT3′, antisense: 5′UAUGCCUCAAGGUCGGUCGUUTT3′) [32] was provided by Sangon Biotech Co., Ltd. (Shanghai, China). Rabbit anti-mouse CD80 antibody, mouse anti-human CD47 antibody, and rabbit anti-human CD80 antibody were obtained from Wuhan Mitaka Biotechnology Co. Ltd. (Wuhan, China). Rabbit anti-human p-PI3K, PI3K, IPS-1, RIG-1, and Bcl-2 antibodies, mouse anti-human p53 antibody, mouse anti-human HLA-1 antibody, and rabbit anti-mouse CD206 antibody were supplied by Abcam. Rabbit anti-human p-ERK, ERK, p-Akt, Akt, and PD-L1 antibodies were supplied by CST. ELISA kits for the detection of mouse IL-12, mouse IL-6, and mouse TNF-α were from 4A Biotech Co., Ltd. (Beijing, China). The ELISA kit for the detection of HBsAg was obtained from Huijia Biotech Co., Ltd. (Xiamen, China).

### 2.2. Cell Lines

HBV genome transfected HepG2.215 cells provided by the National and Local Joint Engineering Research Center of High-Throughput Drug Screening Technology (Wuhan, China) were cultured in DMEM containing 10% fetal bovine serum (FBS) and 100 U/mL penicillin–streptomycin at 37 °C. J774A.1 macrophages obtained from the China Center for Typical Culture Collection (Wuhan, China) were cultured in RPMI 1640 containing 10% FBS and 100 U/mL penicillin–streptomycin at 37 °C. All cells were incubated in a humidified 5% CO_2_ atmosphere.

### 2.3. Preparation and Characterizations of siRNA- and/or pIL-12-Loaded Complexes

siRNA (60 pmol in 50 μL of opti-MEM) was mixed gently with lipofectamine 2000 (2 μg in 50 μL of opti-MEM) followed by incubation for 20 min to form siRNA@lipo complexes.

siRNA (60 pmol) and pIL-12 (1 μg) in opti-MEM (50 μL) were mixed gently with lipofectamine 2000 (2 μg in 50 μL of opti-MEM) followed by incubation for 20 min to form siRNA/pIL-12@lipo complexes.

pIL-12 (1 μg in 50 μL of opti-MEM) was mixed gently with lipofectamine 2000 (2 μg in 50 μL of opti-MEM) followed by incubation for 20 min to form pIL-12@lipo complexes.

siRNA (300 pmol in 100 μL of opti-MEM) was mixed gently with lipofectamine 2000 (4 μg in 100 μL of opti-MEM) followed by incubation for 20 min to form siRNA@lipo* complexes.

siRNA (300 pmol) and pIL-12 (5 μg) in opti-MEM (100 μL) were mixed gently with lipofectamine 2000 (4 μg in 100 μL of opti-MEM) followed by incubation for 20 min to form siRNA/pIL-12@lipo* complexes.

pIL-12 (5 μg in 100 μL of opti-MEM) was mixed gently with lipofectamine 2000 (4 μg in 100 μL of opti-MEM) followed by incubation for 20 min to form pIL-12@lipo* complexes.

The complexes were directly used for further study without removing free siRNA or free pIL-12.

For the measurement of the siRNA and/or pIL-12 encapsulation efficiency, the solution containing complexes was centrifuged at 10,000 rpm for 1 h at 4 °C. After centrifugation, the Quant-iTTM PicoGreen^®^ dsDNA Assay Kit (Molecular Probes) and a spectrofluorophotometer (RF-5301 PC, Shimadzu) were used to determine the amount of unencapsulated free siRNA and/or pIL-12 remaining in the supernatant of solution according to the manufacturer’s protocol. The encapsulation efficiency of siRNA and/or pIL-12 was calculated as follows:encapsulation efficiency = (*W_T_* − *W_F_*)/*W_T_* ×100%,
where *W_F_* is the weight of unencapsulated free siRNA and/or pIL-12, and *W_T_* is the weight of siRNA and/or pIL-12 fed.

The measurement of the size and ζ–potential of complexes, and the procedure for transmission electron microscopy (TEM) observation are detailed in the Appendix A.

### 2.4. In Vitro Cytokine Assay

Cytokines produced by HepG2.215 cells and J774A.1 cells, and HBsAg secreted by HepG2.215 cells before and after being treated with siRNA- and/or pIL-12-loaded complexes for 48 h were detected by enzyme-linked immunosorbent assay (ELISA) as detailed below. A total of 1 × 10^5^ cells were seeded per well of a 12-well plate with 1 mL of culture medium and incubated at 37 °C for 24 h. After that, the medium was replaced with 1 mL of fresh culture medium containing siRNA- and/or pIL-12-loaded complexes. After 48 h, the culture supernatant was collected and then subjected to ELISA according to the manufacturer’s protocol.

### 2.5. Quantitative Polymerase Chain Reaction (qPCR) Assay

HBx mRNA levels in HepG2.215 cells before and after being treated with siRNA- and/or pIL-12-loaded complexes for 48 h were analyzed by qPCR. The details are provided in the Appendix A.

### 2.6. Western Blot Analysis

IPS-1, RIG-1, CD80, HLA-1, PD-L1, CD47, p-PI3K, PI3K, p-Akt, Akt, p-ERK, ERK, Bcl-2, and p53 in HepG2.215 cells, and CD80 and CD206 in J774A.1 cells before and after being treated with siRNA- and/or pIL-12-loaded complexes for 48 h were determined by Western blotting. The details are provided in the Appendix A.

### 2.7. Statistical Analysis

Data are presented as the mean ± standard deviation (SD) based on three independent measurements. The statistical analyses were performed by one-way ANOVA test. Differences were considered statistically significant at *p* < 0.05.

## 3. Results and Discussion

### 3.1. Preparation and Characterization of pIL-12/siRNA-Loaded Complexes

In this study, to achieve combined therapeutic actions of siRNA targeting HBx and IL-12, siRNA and plasmid encoding IL-12 (pIL-12) were coloaded in lipofectamine 2000 (lipo) to form complexes. For comparison, siRNA and pIL-12 were also separately loaded in lipo to study the individual therapeutic effects of siRNA and pIL-12.

Since the sensitivities of hepatocytes (HepG2.215 cells) and immune cells (J774A.1 macrophages) in response to the delivery systems were different, two series of complexes were prepared, i.e., low-dose complexes (pIL-12@lipo, siRNA@lipo, and pIL-12/siRNA@lipo) for HepG2.215 treatments and high-dose complexes (pIL-12@lipo*, siRNA@lipo*, and pIL-12/siRNA@lipo*) for J774A.1 treatment.

The particle size, ζ–potential, morphology, and nucleic acid encapsulation efficiency of the complexes are presented in Figure 1. The particle size (the apparent hydrodynamic diameter of particles determined by light scattering) and ζ–potential of siRNA/pIL-12@lipo were not significantly different from those of siRNA@lipo and pIL-12@lipo, and the particle size (<200 nm) of the complexes was suitable for cellular uptake. Being consistent with the particle size measured by light scattering, transmission electron microscopy (TEM) visualization showed that the complexes containing different nucleic acids were similar in size and spherically shaped.

In this study, lipofectamine 2000 was used as a delivery vector. lipofectamine reagents are considered as the “gold standard” for DNA or RNA delivery. In addition, lipofectamine reagents were also used in anti-hepatitis B-related research. Using lipofectamine as a delivery vector facilitates the comparison of therapeutic efficiencies of our anti-hepatitis B strategy and other anti-hepatitis B strategies. We fabricated the siRNA- and/or pIL-12-loaded complexes according to the instructions of the manufacturer of lipofectamine 2000 to ensure that the complexes did not cause additional cytotoxicity in targeted cells. The ζ–potentials of all complexes were negative. The encapsulation efficiencies of siRNA and pIL-12 were not very high. It should be noted that a further enhancement of the encapsulation efficiency could be achieved by increasing the ratio of lipofectamine 2000 to nucleic acids. However, the increased amount of lipofectamine 2000 would lead to obvious cytotoxicity.

### 3.2. Cytotoxicity of siRNA- and/or pIL-12-Loaded Complexes

In order to ensure that the anti-hepatitis B effects and cytokine secretion were not interfered by the cytotoxicity, we carried out a cytotoxicity study on the vector (lipo) and complexes in HepG2.215 cells and J774A.1 cells. The complex concentrations for cytotoxicity study were kept the same as those for therapeutic siRNA silencing and IL-12 transfection in hepatocytes and macrophages. The proliferation of HepG2.215 cells and J774A.1 cells treated by lipo or diverse complexes did not differ from that of the untreated cells (control) (Figure 2). The good biological safety of therapeutic nucleic acid-loaded complexes ensured the reliability of subsequent studies on the therapeutic actions of these complexes.

### 3.3. Anti-Hepatitis B Effects of siRNA/pIL-12-Loaded Complexes

Before we carried out the study on the codelivery of siRNA and pIL-12, we confirmed the good correlations between the siRNA silencing/pIL-12 transfection results and the concentrations of nucleic acids. The HBsAg level decreased with increasing siRNA concentration (Appendix A). The expression of IL-12 was also strongly dependent on the pIL-12 concentration, i.e., the level of IL-12 increased with increasing pIL-12 concentration (Appendix A). On the basis of these data, we identified the suitable composition for the siRNA/pIL-12@lipo complexes to achieve satisfactory therapeutic outcomes.

After being treated by siRNA@lipo and siRNA/pIL-12@lipo complexes, HBx mRNA levels decreased. Interestingly, pIL-12@lipo complexes also resulted in reduced HBx mRNA.

A previous study showed that HBsAg antigen is abundant in the blood of patients with chronic hepatitis B, and HBsAg may lead to the damage of innate immunity and adaptive immunity, as well as T-cell and B-cell responses. Diverse immune cells have been shown to interact with HBsAg and contribute to the immunopathogenesis of chronic hepatitis B [33], and some guidelines suggest HBsAg seroclearance as an important surrogate marker for complete clearance of HBV [34,35]. In consideration of the critical function of HBsAg in HBV infection, the anti-hepatitis B effects of pIL-12- and/or siRNA-loaded complexes were evaluated on the basis of the expression of HBsAg. Results of ELISA demonstrated that the pIL-12@lipo treatment did not induce HBsAg inhibition, while the siRNA@lipo and the siRNA/pIL-12@lipo treatments significantly reduced the expression of HBsAg (Figure 3). Downregulated HBsAg is beneficial for the regulation and recovery of antivirus immune responses [33].

### 3.4. IL-12 Expression and Reversal of Immnuosuppression in Hepatocytes after Treatment with siRNA- and/or pIL-12-Loaded Complexes

After the treatment for 48 h, both pIL-12@lipo and siRNA/pIL-12@lipo complexes mediated efficient IL-12 expression in HepG2.215 cells (Figure 4). Undoubtedly, the dramatically increased IL-12 in the infected organ would play a crucial role in immunomodulatory to induce antivirus immune responses.

After confirming successful pIL-12 transfection in HepG2.215 cells, we further detected the expression of other immune-related proteins. After the treatment with pIL-12-containing complexes, HepG2.215 cells exhibited enhanced expression of IPS-1, RIG-1, CD80, and HLA-1 and reduced expression of CD-47 and PD-L1 (Figure 5). It should be noted that siRNA/pIL-12-coloaded complexes upregulated IPS-1, RIG-1, CD80, and HLA-1, and downregulated immunosuppressive PD-L1 more significantly as compared with pIL-12-loaded complexes, indicating that siRNA silencing reduced HBx and HBsAg and alleviated immune tolerance, thus favoring the immune regulation of IL-12. Our results are in accordance with a previous report on the transfection of IL-12 promoting expression of CD80 and HLA-1 [36]. Interestingly, siRNA silencing also obviously enhanced the expression of IPS-1 and reduced the expression of PD-L1. This may be due to the silenced HBx protein. Previous studies have shown that there is a certain relationship between HBx and the expression of IPS-1 and PD-L1 [37,38,39], but the specific mechanism needs further study.

As it is well known, IPS-1 and RIG-1 trigger the signaling pathway of type 1 interferon, which is an important part of antiviral activity in innate immunity [40,41]. HBx interacts with IPS-1 and RIG-1 to inhibit the activation of type 1 interferon [39,42]. CD80 activates the T-cell costimulatory receptor CD28 [43,44]. HLA class I (HLA-I) glycoproteins present antigens to cognate CD8^+^ T cells for driving immune responses [45,46]. The upregulation of IPS-1, RIG-1, CD80, and HLA-1 indicates that the codelivery complexes would promote antiviral immune responses. In recent years, studies have shown that the expression of PD-L1 plays an important role in the immunosuppression of patients with chronic hepatitis B. Inhibiting the expression of PD-L1 is beneficial to the recovery of immune function in patients with chronic hepatitis B [47,48,49]. The cluster of differentiation 47 (CD47) can bind with the signal-regulatory protein alpha (SIRP-α) receptor on immune cells to give a signal of “do not eat me” [50,51]. The downregulation of PD-L1 and CD47 demonstrates that codelivery of siRNA and pIL-12 to HBV genome transfected hepatocytes resulted in effective reversal of immunosuppression.

### 3.5. Prevention of HBV-Induced Hepatocellular Carcinoma by siRNA/pIL-12-Loaded Complexes

In view of the important role of HBx in the HBV-induced hepatocellular carcinoma, we also studied the effects of the codelivery system on tumorigenesis-related pathways.

After the treatment with siRNA/pIL-12-loaded complexes for 48 h, the expression of p-PI3K, p-Akt, and p-ERK was downregulated (Figure 6), demonstrating that the PI3K/Akt and ERK-related pathways that promote tumorigenesis were suppressed. In addition, Bcl-2 expression was inhibited, while p53 expression was increased (Figure 6).

Previous studies have shown that HBx induces hepatocellular carcinoma by activating PI3K/Akt and ERK-related pathways, and by regulating the expression of p53 and Bcl-2 [52,53]. The PI3K/Akt pathway is crucial for the progression of hepatocellular carcinoma in terms of proliferation and metastasis [54]. Unusual regulation of the ERK-related pathway leads to abnormal cell growth and proliferation, which promotes carcinogenesis [55]. Bcl-2 functions as an oncogene by blocking apoptosis [56]. On the contrary, the p53 protein is a key tumor suppressor, and loss of p53 function is common during cancer development [57]. Therefore, we believe that the codelivery system can play an important role in hindering the development of HBV-induced hepatocellular carcinoma.

### 3.6. Immune Regulation in Immune Cells by siRNA/pIL-12-Loaded Complexes

The immune regulation efficiency of pIL-12-loaded complexes was studied in J774A.1 macrophages. In our investigation, we used the low-dose complexes (pIL-12@lipo and siRNA/pIL-12@lipo) to treat J774A.1 cells and found that the treatment did not lead to obvious changes in IL-12 level since J774A.1 macrophages were not easily transfected. Thus, we further used the high-dose complexes (pIL-12@lipo* and siRNA/pIL-12@lipo*) for IL-12 transfection. The high-dose complexes could successfully mediate IL-12 expression in J774A.1 cells (Figure 7).

The expression of CD80 increased and the expression of CD206 decreased dramatically in J774A.1 cells after transfection of pIL-12 (Figure 8). Since CD80 is an M1 marker and CD206 is an M2 marker, these results demonstrate that the macrophages differentiated into the antiviral M1 phenotype [21].

In addition, we detected the secretion of IL-6 and TNF-α by J774A.1 cells treated with different complexes. The results show that the concentrations of IL-6 and TNF-α in the culture supernatant of J774A.1 cells significantly increased after transfection of pIL-12 (Figure 9). Since IL-6 and TNF-α are cytokines with direct anti-hepatitis B effects [23], this result confirmed that the pIL-12-containing complexes could effectively induce strong anti-hepatitis B immunity through the secretion of various proinflammatory cytokines with antiviral activities.

The mRNA levels of IL-12, IL-6, TNF-α, CD80, and CD206 in J774A.1 cells after being treated with pIL-12-containing complexes determined by qPCR showed the same trend (Appendix A), further confirming the effective immune regulation.

### 3.7. Overview of Immunomodulatory Effects of siRNA/pIL-12-Loaded Complexes

As detailed above, the codelivery system not only inhibited HBV but also effectively reversed the immnuosuppression of HBV in hepatocytes (HepG2.215 cells) (Figure 1). The immunosuppression caused by hepatitis B virus was successfully reversed, i.e., immune-stimulatory IPS-1, RIG-1, CD80, and HLA-1 were upregulated and immune-suppressive PD-L1 and CD47 were effectively downregulated in hepatocytes. Furthermore, the codelivery system inhibited the PI3K/Akt and ERK pathways in hepatocytes and upregulated p53 expression, alleviating the risk of HBV-induced tumorigenesis. Moreover, the macrophages treated by the codelivery system were successfully differentiated into the M1 phenotype, expressing cytokines with anti-hepatitis B effects.

According to previous studies, the delivery of siRNA or 5′-triphosphate siRNA targeting HBx showed effective anti-HBV effects and the potential of immune activation [58,59], and the delivery of pIL-12 resulted in enhancing HBV-specific CD8^+^ and CD4^+^ T-cell responses [60,61]. As far as we know, no reversal of immunosuppression by either siRNA or pIL-12 has been reported in hepatitis B treatments. Our codelivery systems can efficiently reverse HBV-induced immunosuppression and prevent HBV-induced hepatocellular carcinoma. This study provides a promising strategy to overcome the hurdles in hepatitis B therapy.

## 4. Conclusions

The codelivery system of an siRNA targeting HBx and plasmid encoding IL-12 was prepared for HBV treatments to achieve the enhanced antiviral efficiency. The treatment with siRNA/pIL-12-coloaded complexes could effectively knock down HBx and result in markedly reduced HBsAg in HBV genome transfected HepG2.215 cells. More importantly, as compared with pIL-12-loaded complexes, siRNA/pIL-12-coloaded complexes exhibited enhanced capability in downregulating immunosuppressive PD-L1 and upregulating immune-boosting IPS-1, RIG-1, CD80, and HLA-1. Clearly, the decreased HBsAg load alleviated immune tolerance and favored immune activation. Furthermore, the siRNA/pIL-12 codelivery system exhibited good potential in preventing HBV-induced hepatocellular carcinoma by suppressing the PI3K/Akt and ERK pathways. In addition, the transfection of pIL-12 in J774A.1 macrophages mediated by the siRNA/pIL-12-loaded complexes led to successful immunoregulation and shifted the phenotype of macrophages to the M1 type with increased secretion of antiviral cytokines such as IL-6 and TNF-α. Therefore, the codelivery of siRNA and pIL-12 can optimize the therapeutic efficiency via synergistic effects of inhibition of HBV, reversal of immunosuppression, reactivation of antiviral immunity, and prevention of HBV-induced hepatocellular carcinoma.

## Data Availability

The data presented in this study are available on request from the corresponding author.

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
