# Peer review of "Codelivery of HBx-siRNA and Plasmid Encoding IL-12 for Inhibition of Hepatitis B Virus and Reactivation of Antiviral Immunity"

_pharmaceutics, 2022, doi:10.3390/pharmaceutics14071439_

Round 1
Reviewer 1 Report
The manuscript pharmaceutics-1782347 Co-delivery of HBx-siRNA and Plasmid Encoding IL-12 for Inhibition of Hepatitis B Virus and Reactivation of Antiviral Immunity by Yan Mu et al. describes the study of siRNA and pIL-12 delivery systems to improve hepatitis B virus inhibition.
The manuscript is logical and well written. The prospects for the practical use of the results obtained are high. The paper will definitely be of interest to the readers of Pharmaceutics.
Questions and comments:
1) Lipofectamine is widely used as a vector for gene delivery. What is the novelty of your research? What lipofectamine-based delivery systems for hepatitis virus inhibition were described in the literature and what are their drawbacks compared to your polyplexes?
2) In section 2.3. the description of the methodology of complex preparation should be optimized. Perhaps the initial conditions for preparation should be summarized in a table, where the ratios (molar or mass) between the components should be specified.
3) The term "Zeta potential" as a characteristic of the surface charge of the polyplex is written differently in the text of the manuscript. Perhaps, for clarity, it should be designated as ζ-potential.
4) It should also be explained in the text that the particle size in this case is their apparent hydrodynamic diameter.
5) Section 3.2 - Why were the cytotoxicity data for pure lipofectamine not presented in Fig. 2?
6) The methodology of determining the encapsulation efficiency should be presented in the text of the article.
Author Response
1) Lipofectamine is widely used as a vector for gene delivery. What is the novelty of your research? What lipofectamine-based delivery systems for hepatitis virus inhibition were described in the literature and what are their drawbacks compared to your polyplexes?
Answer: Although lipofectamine-based delivery systems have been investigated as the delivery vector for nucleic acids with anti-HBV effects [Antiviral Research. 2015 doi: 10.1016/j.antiviral.2015.03.015; Theranostics. 2017 doi: 10.7150/thno.18114; Antiviral Research. 2018 doi: 10.1016/j.antiviral.2018.02.011], no studies on the reversal of immune tolerance have been reported. For anti-HBV therapy, reversing immune tolerance and activating antiviral immunity are crucial for the body to maintain a clear state of HBV after drug withdrawal. Thus, the purpose of our study is to reverse immune tolerance and activate antiviral immunity by the co-delivery system.
The main innovation of our study is the immunomodulatory effects of co-delivery of siRNA and pIL-12 for anti-hepatitis B purpose, which have never been reported. Our study indicates that co-delivery of siRNA and pIL-12 can effectively reverse immune tolerance induced by hepatitis B virus, which is of importance for the treatment of chronic hepatitis B. Furthermore, the siRNA/pIL-12 co-delivery system exhibits good potential in hindering the progression of HBV-induced hepatocellular carcinoma.
2) In section 2.3. the description of the methodology of complex preparation should be optimized. Perhaps the initial conditions for preparation should be summarized in a table, where the ratios (molar or mass) between the components should be specified.
Answer: As suggested, the conditions and the ratios between the components for the complex preparation are summarized and added in the Supporting Information as Table S1.
3) The term "Zeta potential" as a characteristic of the surface charge of the polyplex is written differently in the text of the manuscript. Perhaps, for clarity, it should be designated as ζ-potential.
Answer: As suggested, "Zeta potential" was replaced by "ζ-potential".
4) It should also be explained in the text that the particle size in this case is their apparent hydrodynamic diameter.
Answer: As suggested, the explanation on the particle size (the apparent hydrodynamic diameter) was added in the main text.
5) Section 3.2 - Why were the cytotoxicity data for pure lipofectamine not presented in Fig. 2?
Answer: As suggested, the cytotoxicity data for pure lipofectamine was studied and the result was added in Fig. 2.
6) The methodology of determining the encapsulation efficiency should be presented in the text of the article.
Answer: As suggested, the methodology of determining the encapsulation efficiency is moved to the main text (2.3. Preparation and Characterizations of siRNA and/or pIL-12 Loaded Complexes.).

Reviewer 2 Report
The manuscript introduced biological evaluation of pIL-12/siRNA-loaded complexes. The authors showed that the co-delivery of siRNA and pIL-12 was able to optimize the therapeutic efficiency via synergistic effects of inhibition of HBV, reversal of immunosuppression, reactivation of antiviral immunity and prevention of the HBV-induced hepatocellular carcinoma. Thus, these findings will be useful for the treatment of chronic hepatitis B. Therefore, the manuscript is not too excellent to be published. In other words, the manuscript is so excellent that it should be published.
Comments
(1) Was the plasmid encoding interleukin 12 (pIL-12) introduced into cells other than hepatocytes and immune cells?
(2) Were the siRNA and pIL-12 enzymatically stable owing to ipofectamine 2000, particularly, in the serum?
(3) In what mechanism was the internalization of siRNA and pIL-12 loaded on ipofectamine 2000 into hepatocytes and immune cells? Endocytosis? Fusion?
That is all.
Author Response
1) Was the plasmid encoding interleukin 12 (pIL-12) introduced into cells other than hepatocytes and immune cells?
Answer: Yes, the co-delivery complex may be uptaken by other cells. Nevertheless, since the particle size is in the range of 100 to 200 nm, the complexes loaded with pIL12 are easily phagocytosed by macrophages and accumulated in liver.
2) Were the siRNA and pIL-12 enzymatically stable owing to lipofectamine 2000, particularly, in the serum?
Answer: Yes, they are stable owing to lipofectamine 2000. The medium used in our experiments contains 10% fetal bovine serum, and the complexes loaded with siRNA and pIL-12 can deliver siRNA and pIL-12 into living cells to mediate effective siRNA silencing and pIL-12 transfection, implying that siRNA and pIL-12 encapsulated in lipofectamine 2000 are enzymatically stable before cellular uptake.
3) In what mechanism was the internalization of siRNA and pIL-12 loaded on lipofectamine 2000 into hepatocytes and immune cells? Endocytosis? Fusion?
Answer: According to previous reports, the combined transmembrane routes through the clathrin and caveolae-mediated pathways are the major mechanisms of cell uptake for the lipofectamine 2000-mediated gene delivery [Biotechnol Lett. 2014 doi: 10.1007/s10529-013-1325-0]. Since we used lipofectamine 2000 as the delivery vector, the internalization mechanism of siRNA and pIL-12 loaded complexes should be clathrin and caveolae-mediated pathways.

Round 2
Reviewer 1 Report
The manuscript may be accepted for publication
This manuscript is a resubmission of an earlier submission. The following is a list of the peer review reports and author responses from that submission.
Round 1
Reviewer 1 Report
The manuscript entitled ‘Co-delivery of siRNA and Plasmid Encoding IL-12 for Inhibition
of Hepatitis B Virus and Reactivation of Antiviral Immunity’ by Mu et al. describes the
co-delivery of siRNA and plasmid IL-12 in macrophages (J774) and Hepatitis infected
liver cells as possible therapeutic for HBV treatment. The study shows that co-delivery
of siRNA targeting HBx protein of virus and plasmid IL-12 has the potential to treat HBV
by giving a boost to patient immune system. Specifically, due to IL-12 transfection M0
macrophages can be polarized to M1 hence, combatting viral infection. This is
supported by ELISA assays, showing the production of pro-inflammatory cytokines and
upregulation of pro-inflammatory proteins by western blot. Although the analysis seems
to be performed by RT-PCR as well, but the data is missing.
Overall, this is a neat idea but is in very preliminary stages and the data provided is not
sufficient for publication.
First the manuscript lacks novelty. Use of lipofectamine for transfection is not ideal, and
is not expected to be translated to any practical applications. Furthermore, this is
incorrect to say that this is the first study focusing on co-delivery of siRNA and pDNA,
there are other studies, they may not be focused on HBV treatment.
There are other critical errors in the manuscript. The transfection efficacies/treatments
are only reliable if performed on range of concentrations of nucleic acids, as this
provides a true overview of therapeutic effect and must be dose dependent. Taking one
concentration of siRNA and DNA and showing their effect is not ideal and the data
provided may have originated from any off target effects or other discrepancies.
This is unclear why J774 cells were used for the study. This cell line is of mouse origin,
while HepG2 cells are of human origin.
Author Response
Answers to comments from Reviewer 1
The manuscript entitled ‘Co-delivery of siRNA and Plasmid Encoding IL-12 for Inhibition of Hepatitis B Virus and Reactivation of Antiviral Immunity’ by Mu et al. describes the co-delivery of siRNA and plasmid IL-12 in macrophages (J774) and Hepatitis infected liver cells as possible therapeutic for HBV treatment. The study shows that co-delivery of siRNA targeting HBx protein of virus and plasmid IL-12 has the potential to treat HBV by giving a boost to patient immune system. Specifically, due to IL-12 transfection M0 macrophages can be polarized to M1 hence, combatting viral infection. This is supported by ELISA assays, showing the production of pro-inflammatory cytokines and upregulation of pro-inflammatory proteins by western blot. Although the analysis seems to be performed by RT-PCR as well, but the data is missing.
Answer: As required, additional experiments were carried out. The corresponding pro-inflammatory cytokines and proteins were detected by RT-PCR, and the data were added in the Supporting Information as Figure S3.
Overall, this is a neat idea but is in very preliminary stages and the data provided is not sufficient for publication.
Answer: As required, additional experiments were carried out and the results were added in the revised manuscript as detailed in the answers below.
First the manuscript lacks novelty. Use of lipofectamine for transfection is not ideal, and is not expected to be translated to any practical applications. Furthermore, this is incorrect to say that this is the first study focusing on co-delivery of siRNA and pDNA, there are other studies, they may not be focused on HBV treatment.
Answer: The main purpose of our study is to develop an anti-hepatitis B strategy instead of a novel gene delivery vector. The reason we used lipofectamine as a delivery vector is that lipofectamine reagents are considered as a “gold standard” for DNA or RNA delivery [Sci Rep. 2016 doi: 10.1038/srep25879]. In addition, lipofectamine reagents were also used in anti-hepatitis B related research [Theranostics 2017, 22, 3090; Theranostics 2020, 23, 9249; J Cell Mol. Med. 2019, 23, 5920]. As a results, using lipofectamine as a delivery vector facilitates the comparison of therapeutic efficiencies of our anti-hepatitis B strategy and other anti-hepatitis B strategies. So, we selected lipofectamine as a vector.
The main innovation of our study is the immunomodulatory effects of co-delivery of siRNA and pIL-12 for anti-hepatitis B purpose, which have never been reported. Our study indicates that co-delivery of siRNA and pIL-12 can effectively reverse immune tolerance induced by hepatitis B virus, which is of importance for the treatment of chronic hepatitis B.
Yes, the studies on co-delivery of siRNA and pDNA for cancer treatments have been reported. However, there is no report on co-delivery of siRNA and pDNA encoding IL-12 for hepatitis B treatments. This novelty of our study is emphasized in the Introduction section.
There are other critical errors in the manuscript. The transfection efficacies/treatments are only reliable if performed on range of concentrations of nucleic acids, as this provides a true overview of therapeutic effect and must be dose dependent. Taking one concentration of siRNA and DNA and showing their effect is not ideal and the data provided may have originated from any off target effects or other discrepancies. This is unclear why J774 cells were used for the study. This cell line is of mouse origin, while HepG2 cells are of human origin.
Answer: We are sorry that we only report the optimized concentrations in the original version. As required, additional data on the dependence of transfection efficacies/treatments on the concentrations of nucleic acids were added in the Supporting Information as Figure S1 and Figure S2.
The HBsAg level decreases with increasing siRNA concentration (Figure S1). The expression of IL-12 is also strongly dependent on the pIL-12 concentration, i.e., the level of IL-12 increases with increasing pIL-12 concentration (Figure S2). The good correlations between the siRNA silencing/pIL-12 transfection results and the concentrations of nucleic acids implies that the result is a true therapeutic effect instead of off target effects or other discrepancies.
Since pIL-12 used in our study encoding mouse IL-12, we chose J774A.1 cells as the targeted cells. (Human and mouse IL-12 has 70% and 60% homology of the p40 and p35 subunits, respectively.) The data obtained using mouse IL-12 would facilitate in vivo studies on mouse models in the further. (Mouse models are commonly used in in vivo research of anti-hepatitis B.)

Reviewer 2 Report
The manuscript by Mu et al. represents a decent study on lipofection of HepG2.215 cells and co-delivery of two therapeutic agents. The article is quite well designed, however, there are some concerns to be considered before publication.
Minor remarks:
1/ Materials and Methods - the description of statistical analyses methodology should be completed in this chapter.
2/ Lines 143-148 - the Authors describe their final outcomes at the very beginning of "Results" section. This part should be placed after presenting results.
3/ Scheme 1 presents a summary of obtained results, it would be clearer to place it at the end of article.
4/ The PD-L1 abbreviation should be explained when used for the first time in text.
5/ The Authors barely discuss their results with the literature data. There is no "Discussion" chapter at all.
6/ English revision of the manuscript is absolutely required before its publication.
Author Response
Answers to comments from Reviewer 2
The manuscript by Mu et al. represents a decent study on lipofection of HepG2.215 cells and co-delivery of two therapeutic agents. The article is quite well designed, however, there are some concerns to be considered before publication.
Minor remarks:
1/ Materials and Methods - the description of statistical analyses methodology should be completed in this chapter.
Answer: As suggested, “Statistical analysis” was added in the “Materials and methods” section.
2/ Lines 143-148 - the Authors describe their final outcomes at the very beginning of "Results" section. This part should be placed after presenting results.
Answer: As suggested, final outcomes is placed after presenting results.
3/ Scheme 1 presents a summary of obtained results, it would be clearer to place it at the end of article.
Answer: As suggested, Scheme 1 and the related text are placed at the end of article.
4/ The PD-L1 abbreviation should be explained when used for the first time in text.
Answer: As suggested, PD-L1 abbreviation is defined at the first appearance in “Abstract”.
5/ The Authors barely discuss their results with the literature data. There is no "Discussion" chapter at all.
Answer: We are sorry for missing “Discussion”, “Results” section was revised to “Results and Discussion”. More discussion on our results and the comparison on our results and literature results (cited as Refs. 48-51) were added in this section.
6/ English revision of the manuscript is absolutely required before its publication.
Answer: As suggested, the language is carefully revised.

Reviewer 3 Report
This manuscript incorporates two bioactive molecules (RNA and pDNA) in one nanostructure. This theme is of relevant significance. However, there are major issues to be answered before publication. Are there other papers that report this strategy for gene delivery?
Why do the authors decide to use lipofectamine? This selection should be clear in the manuscript. Lipofectamine is used in the laboratory only. Then, in my understanding, it is a proof of concept. The authors should make it clear.
The size method should be explained in the methods section. What equipment was used? Did the authors use z-average? What was the polydispersity index? This is an important issue that can help in understanding the biological effect.
Method for zeta potential, TEM, and encapsulation efficiency should also be described.
The encapsulation efficiency is for both RNA and DNA? It is not clear. If 50-60% encapsulation is reached, and considering the drive-force to retain genetic material is the cationic lipid, the authors should better discuss what happened. The proportion of lipofectamine is not proportional to the genetic material added?
If the encapsulation efficiency is low, the zeta potential results are a consequence of the low amount of cationic lipid. The genetic material excess is lost or free in the formulation.
The unencapsulated material is removed from the formulation? This is an important issue, and the authors have to clarify it.
The transfection method is not described. How long does it take to change the culture media after adding the lipid/DNA/RNA complexes to the cell culture in plates? The characterization of protein IL-12 Expression and Reversal of Immunosuppression in Hepatocytes occur 48h after changing the culture media?
It seems there is no considerable difference in using both DNA and RNA. The authors should discuss this result relating to the encapsulation efficiency.
Discussion should be improved and comparison with other DNA/RNA encapsulation investigations.
The supplementary material is not accessible.
Author Response
Answers to comments from Reviewer 3
This manuscript incorporates two bioactive molecules (RNA and pDNA) in one nanostructure. This theme is of relevant significance. However, there are major issues to be answered before publication. Are there other papers that report this strategy for gene delivery?
Answer: To our best knowledge, there is no reports on co-delivery of siRNA and pDNA encoding IL-12 for hepatitis B treatments. This explanation on our novelty was added in the Introduction section.
Why do the authors decide to use lipofectamine? This selection should be clear in the manuscript. Lipofectamine is used in the laboratory only. Then, in my understanding, it is a proof of concept. The authors should make it clear.
Answer: The reason we used lipofectamine as a delivery vector is that lipofectamine reagents are considered as a “gold standard” for DNA or RNA delivery [Sci Rep. 2016 doi: 10.1038/srep25879]. In addition, lipofectamine reagents were also used in anti-hepatitis B related research [Theranostics 2017, 22, 3090; Theranostics 2020, 23, 9249; J Cell Mol. Med. 2019, 23, 5920]. As a results, using lipofectamine as a delivery vector facilitates the comparison of therapeutic efficiencies of our anti-hepatitis B strategy and other anti-hepatitis B strategies. So, we selected lipofectamine as a vector.
The above explanation was added in the revised manuscript.
The size method should be explained in the methods section. What equipment was used? Did the authors use z-average? What was the polydispersity index? This is an important issue that can help in understanding the biological effect.
Answer: As suggested, the detailed experimental procedure on the measurements of particle size, and PDI are provided in Supplementary Methods in SI. Yes, we used z-average.
Method for zeta potential, TEM, and encapsulation efficiency should also be described.
Answer: As suggested, the detailed experimental procedure on the measurements of zeta potential and encapsulation efficiency, and the TEM characterization are provided in Supplementary Methods in SI.
The encapsulation efficiency is for both RNA and DNA? It is not clear. If 50-60% encapsulation is reached, and considering the drive-force to retain genetic material is the cationic lipid, the authors should better discuss what happened. The proportion of lipofectamine is not proportional to the genetic material added?
Answer: For the co-delivery system, the encapsulation efficiency is for both RNA and DNA.
We fully agree that the encapsulation efficiency is not very high. In fact, we tried to enhance the encapsulation efficiency by increasing the lipofectamine to siRNA/pIL-12 ratio. However, the increased amount of lipofectamine leads to obvious cytotoxicity. To ensure that lipofectamine does not bring obvious cytotoxicity, we used the formulation reported in the manuscript. In addition, the ratio of cationic liposomes to nucleic acids in the current manuscript is also recommended by the manufacturer.
If the encapsulation efficiency is low, the zeta potential results are a consequence of the low amount of cationic lipid. The genetic material excess is lost or free in the formulation.
Answer: Yes, there are free nucleic acids exist in the formulation.
The unencapsulated material is removed from the formulation? This is an important issue, and the authors have to clarify it.
Answer: The unencapsulated material was not removed in the current experiment. In fact, we tried to remove unencapsulated nucleic acids, but it is very difficult to completely remove them by centrifugation or dialysis. This is clarified in the revised manuscript.
The transfection method is not described. How long does it take to change the culture media after adding the lipid/DNA/RNA complexes to the cell culture in plates? The characterization of protein IL-12 Expression and Reversal of Immunosuppression in Hepatocytes occur 48h after changing the culture media?
Answer: As suggested, the transfection method is described in detail in the Supporting Information. The culture media were not changed after adding the lipid/DNA/RNA complexes, and the cells were co-incubated with the lipid/DNA/RNA complexes for 48 h.
Before characterization of protein IL-12 expression by ELISA, the culture media were not removed.
Before the study on the reversal of Immunosuppression by PCR and Western blotting, the culture media were removed.
These experimental details were added in the revised manuscript.
It seems there is no considerable difference in using both DNA and RNA. The authors should discuss this result relating to the encapsulation efficiency.
Answer: As suggested, the discussion on the result relating to the encapsulation efficiency was added in “Results and discussion”.
Discussion should be improved and comparison with other DNA/RNA encapsulation investigations.
Answer: As suggested, the detailed discussion on the result relating to the encapsulation efficiency was added to improve the Discussion.
The supplementary material is not accessible.
Answer: To ensure the supplementary material is accessible, we combined the supplementary material with the main text into one file.

Round 2
Reviewer 1 Report
Thank you for sending the revised version and addressing some comments.
I am still not convinced about the novelty of this study. The method used here has no relevance to practical applications, as lipofectamine can be used for treatment of humans.
Furthermore the results presentation and discussion is extremely unclear and lacks precision.
Pharmaceutics is a well reputed journal in the field with impact factor of 6.3; given the scope and presentation of this study, I would suggest to explore other journals for this study.
Reviewer 3 Report
The authors answered all questions. The manuscript can be accepted in the present form.